

# Image reconstruction in graphic design based on Global residual Network optimized compressed sensing model

Xinxin Fu[1], Lujing Tang[1] and Yingjie Bai[2]

[1] Department of Integrated Industrial Design, Hanseo University, Seosan, Republic of South Korea
[2] School of Design, Guangxi Normal University, Guilin, China

## ABSTRACT

The article aims to address the challenges of information degradation and distortion in graphic design, focusing on optimizing the traditional compressed sensing (CS) model. This optimization involves creating a co-reconstruction group derived from compressed observations of local image blocks. Following an initial reconstruction of compressed observations within similar groups, an initially reconstructed image block co-reconstruction group is obtained, featuring degraded reconstructed images. These images undergo channel stitching and are input into a global residual network. This network is composed of a non-local feature adaptive interaction module stacked with the aim of fusion to enhance local feature reconstruction. Results indicate that the solution space constraint for reconstructed images is achieved at a low sampling rate. Moreover, high-frequency information within the images is effectively reconstructed, improving image reconstruction accuracy.

# INTRODUCTION

With the advent of the Internet and self-media, the demand for graphic design has significantly increased. The design requirements for image quality and clarity have proportionally escalated, reflecting the evolution of the digital age (*Li et al., 2020a*). Artificial intelligence (AI) algorithms can perform edge detection, recognition, matching, segmentation, and image classification of images, thereby optimizing function and enabling fuzzy control in the automatic production process of digital images. The integration of AI in computer graphic design has facilitated the realization of the automatic generation process of intelligent images (*Gavini & Lakshmi, 2022*; *Nehashree, 2019*; *Jing et al., 2022*). In computer graphic design, primary attention should be given to the linguistic expression of information design. Graphic symbols are a unique means of conveying properties of objects, which is unparalleled by other representations. However, external factors, such as noise during the acquisition and transmission, may distort graphic images. High-resolution images contain intricate details crucial for creating professional and visually appealing designs. When information is degraded, these fine details can be lost, resulting in images that appear blurry, pixelated, or otherwise of poor quality. This loss of clarity can diminish the

Corresponding author
Lujing Tang, 18055406588@163.com

impact of the design and reduce its effectiveness in communicating the intended message. Edges and contours are vital elements in defining the shapes and boundaries within an image. Distortion in these areas can lead to jagged or blurred edges, compromising the visual integrity of the design. This is particularly problematic in applications requiring precise and clean lines, such as logos, typography, and detailed illustrations. As a result, the restoration of large-scale distorted images mandates low-complexity methods that can accurately reconstruct the images (*Lan et al., 2020*; *Aburukba et al., 2020*).

The elemental image matrices utilized in graphic design are stored as pixel points, resulting in often sparse matrices. The application of the compressed sensing (CS) algorithm allows for high-dimensional graphic matrices to be projected onto low-dimensional space, where they can be optimally resolved using matching algorithms. The distorted image reconstruction can be achieved by reconstructing the original signal matrix, utilizing the low-dimensional observed signal matrix. CS theory (*Shi et al., 2019*; *Wang et al., 2023*; *Xu et al., 2020*) demonstrates that a signal sparsely represented on an appropriate transform base can be recovered with fewer sampled measurements than those required by the conventional Nyquist sampling theorem.

Within the CS framework, the simultaneous occurrence of sampling and compression offers significant advantages, including reduced bandwidth requirements for data transmission and lower memory usage for storing compressed image signals. In the context of planar design, the image is divided into multiple non-overlapping image blocks, which are subsequently sampled and independently reconstructed. This approach facilitates the equitable distribution of perceptual resources across the image, thanks to the uneven distribution of meaningful information (*Shi et al., 2019*). However, the use of block-based CS comes with a drawback known as the "block effect", which can potentially diminish the quality of image reconstruction.

CS utilizes signal sparsity in the transform domain to perform simultaneous signal sampling and compression. Signal reconstruction, a key difficulty in CS, involves effectively recovering information from the obtained compressed observations using the corresponding reconstruction algorithm. However, the stability of the reconstruction process, accuracy of the reconstructed signal, and improvement of reconstruction accuracy are essential components of the CS reconstruction problem that require in-depth study (*Li et al., 2020b*; *Sun et al., 2020*). In traditional CS reconstruction problems, hundreds of iterations are utilized to optimize the reconstructed signal continuously. However, this approach can result in information loss accumulation, increased computational complexity, and hardware device overload, thus hindering the widespread use of CS techniques in image reconstruction problems (*Stanković, Orović & Stanković, 2014*).

Furthermore, as an undefined underdetermined problem, CS information reconstruction can cause instability and uncertainty during signal reconstruction. Insufficient CS sampling information can result in blurred and information issues of loss in the reconstructed image (*Huang & Wang, 2015*). Since the emergence of deep learning techniques, the field of image information has rapidly developed, and combining deep learning techniques with reconstructive vision tasks such as image restoration and

image super-resolution has shown significant potential. Consequently, applying deep learning techniques to CS image reconstruction problems is becoming a mainstream trend.

Given the limited structural information contained within an image and the prevalence of repetitive and similar features within these structures, it becomes necessary to employ non-local similar features to provide more effective information for local image reconstruction. This approach is particularly useful when local image reconstruction information is insufficient. (1) This article proposes a novel image reconstruction model that utilizes a non-local feature fusion network. A collaborative image block reconstruction group is introduced in the initial image reconstruction stage. (2) The model employs full convolutional residual blocks to avoid deepening the network structure, which can lead to the loss of image features in the forward channel and impede network convergence.

The article begins by summarizing the current state of research on image CS and introducing the structure and general framework of the designed model. The application of the residual network in CS image reconstruction is also explored. Finally, a series of experimental comparisons are conducted to demonstrate the effectiveness of the proposed network.

## RELATED WORKS

Compared to the intricate modeling and optimization process of CS reformulation, deep learning can approximate arbitrarily complex functions using deep nonlinear network structures. This approach can learn autonomously from big data to fit a network model to the problem, resulting in more efficient reconstruction than traditional algorithms (*Zhou et al., 2020*). Combining CS and neural network nonlinear reconstruction can solve the problems inherent in traditional CS theory using deep learning models (*Luo, Liang & Ren, 2022*). Deep neural networks (DNNs) have demonstrated their superiority over other solutions and have been widely used in many computer vision tasks in recent years (*Darestani, Liu & Heckel, 2022*; *Bian, Cao & Mao, 2022*; *Chai et al., 2022*). Several deep neural network-based image CS methods have been proposed, with ReconNet, a convolutional neural network developed by *Kulkarni et al. (2016)*, one of the early examples. *Yao et al. (2019)* later proposed DR2-Net, which improved the image reconstruction quality compared to ReconNet by utilizing a residual network (*He et al., 2016*) structure based on the ReconNet network. *Zhang & Ghanem (2018)* developed ISTA-Net, a new structured deep network inspired by the traditional iterative shrinkage thresholding algorithm, which maintains good mathematical interpretability in deep networks.

Additionally, *Zhou, Liu & Shen (2023)* proposed an optimization-inspired attentional neural network for deep CS, which implements the fusion of approximate message-passing algorithms with neural networks and introduces the attentional network structure. *Kumar et al. (2020)* proposed CSNet, an end-to-end network framework that uses a convolutional layer to model the sampling matrix and convolutional networks for CS recovery. The network performs joint optimization of sampling and reconstruction parameters and obtains good reconstruction results. To effectively preserve the texture details of images,

*Sun et al. (2020)* proposed a dual-path attention network for CS image reconstruction, which employs structure path, texture path, and texture attention modules to reconstruct the image structure and texture components. The network-based CS method offers joint training of sampling and recovery phases and is non-iterative, significantly reducing the time complexity compared to optimization-based approaches.

The blocking effect refers to the visual artifacts that arise when adjacent image blocks are treated independently during reconstruction. Since each block is processed individually, discontinuities and inconsistencies can appear at the boundaries where the blocks meet without considering the context of neighboring blocks. These artifacts disrupt the overall visual coherence and fidelity of the reconstructed image. Despite this drawback, several techniques have been developed to mitigate the block effect in block-based CS. One common approach involves employing overlapping blocks during the sampling and reconstruction stages. By allowing adjacent blocks to overlap, the context and correlations between neighboring blocks are considered, reducing the occurrence of visible artifacts at block boundaries. Furthermore, advanced algorithms and optimization techniques have been devised to improve the accuracy and quality of image reconstruction in block-based CS. These methods aim to enhance the representation of image features within each block and ensure smooth transitions between adjacent blocks.

In summary, block-based CS offers benefits such as equitable resource distribution and efficient compression, but the block effect remains a challenge that can impact image reconstruction quality. Nevertheless, ongoing research and advancements in overlapping block techniques and optimization algorithms continue to address this issue, paving the way for improved block-based CS approaches with higher fidelity and visual coherence in reconstructed images.

Current image compression-aware reconstruction techniques utilize the compressed sensory observations' sampling method to distinguish and construct different reconstruction methods, achieving high accuracy and effective restoration of image information. However, these methods do not utilize the image structure a priori information to guide the design of the deep network framework, which can achieve optimal constraints on the image solution space and effectively refine the image structure's sparse a priori information to achieve accurate reconstruction of images.

## METHODS

Graphic design work is predominantly accomplished with the aid of specialized software (Adobe Photoshop, Adobe Illustrator, CorelDRAW, Affinity Designer, Sketch that enables the completion of complex two-dimensional spatial design and arrangement, encompassing color and saturation adjustments to achieve the desired effect. It is crucial to note that the designed work must be aesthetically pleasing to appeal to a wider audience. As an art form, graphic design encompasses various artistic elements, and it must effectively convey the design theme while serving as a tool for publicity and inspiration. The indispensable support of computer technology enables efficient graphic design and image processing, but noise and information loss inevitably occur, resulting in image distortion. To rectify these

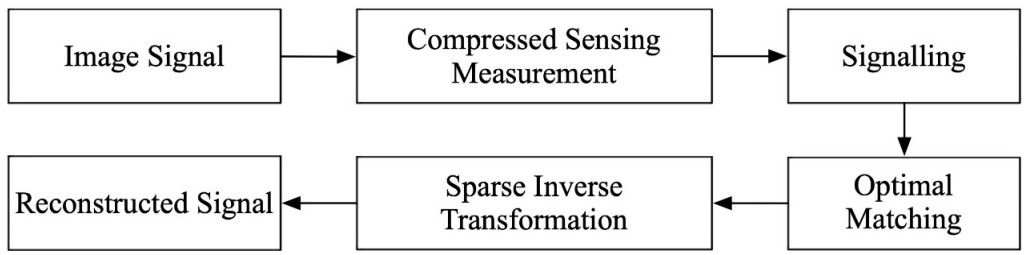

**Figure 1  CS signal reconstruction model flow.**

issues, the CS method can be employed to denoise and reconstruct distorted images. The CS algorithm projects the high-dimensional graphic matrix into a low-dimensional space and optimizes it by matching the optimal algorithm. This process reconstructs the original signal matrix using the low-dimensional observed signal matrix and ultimately achieves the desired reconstruction of the distorted image, as illustrated in Fig. 1.

The optimal approximation of the sparse graphical matrix is achieved by matching the prior sequence with the observation matrix, ultimately resulting in image reconstruction and noise reduction.

## Overall design

Drawing inspiration from non-local mean value theory and leveraging deep learning technology, the author proposes an image CS reconstruction method based on non-local feature fusion network. The model is depicted in Fig. 2. To enhance the reconstruction of image block information, this article adopts an adaptive fusion of non-local a priori information using the image non-local mean theory, thereby enabling a cooperative representation of image information. The image structure information is constructed by designing a two-stage reconstruction network that progressively refines the image from coarse to fine.

During the enhancement reconstruction stage, the image non-local similarity prior plays a crucial role in providing complementary information to reconstruct individual image block features, thereby fully utilizing the internal structural features of the image. To this end, a non-local feature adaptive interaction module is designed, comprising two non-local feature fusion convolutions, a channel correlation discriminator module, and a spatial correlation discriminator module. This adaptive weighted fusion of non-local image features enables the suppression of mismatching information.

## Collaborative reconfiguration group construction

The original image is first segmented into B × B blocks of non-overlapping size $x_i, i = 1, 2, \ldots, N$ and then the blocks are transformed into $B^2 \times 1$ dimensional column vector, and use Gaussian random sampling matrix $\phi$ to obtain the compressed sampling values. $y_i$, whose observation matrix is $\Phi$ , then the signal measurement model can be expressed as Eq. (1).

$$y_i = \Phi_{x_i}. \tag{1}$$

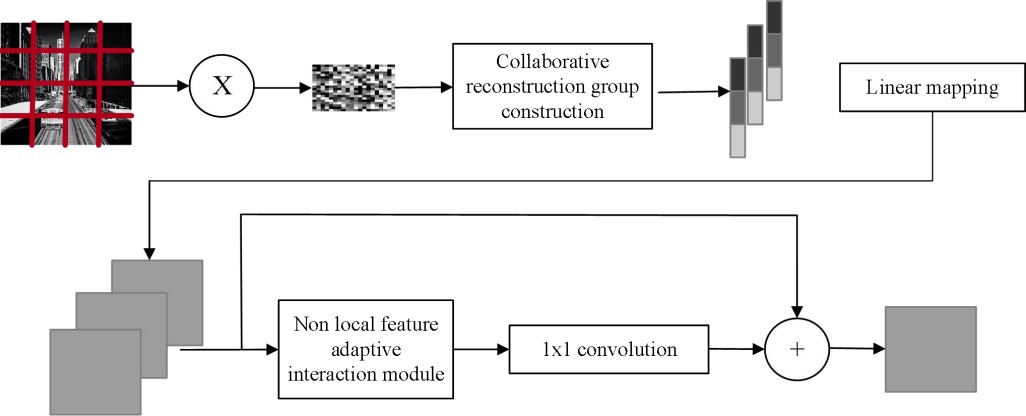

**Figure 2  Model structure.**

When the observation matrix and the measurement results are known, the original graph matrix can be recovered using a one-dimensional parametric definition.

$$x = \min \|x\|_1 \tag{2}$$

s.t. $y_i = \Phi x_i$.

Take B size 16, convert the image block to a $B^2$ dimensional column vectors, using the observation matrix $\phi$ for sampling.

The cosine similarity metric effectively captures the internal structural changes of an image while minimizing the impact of uneven illumination on the construction of similar blocks when measuring the similarity of two image blocks. Therefore, in this article, the cosine similarity of the compressed observations of the two blocks is computed to assess the similarity of their observations. By comparing the similarity of the compressed observations of a single image block with those of all other image blocks, the author ranks the obtained cosine similarity values from the largest to the smallest, selecting the compressed observations with greater similarity to form a cooperative reconstruction group $Y_i$.

The linear mapping network F was used to upgrade the dimension of the cooperative reconstruction group $Y_i$ to obtain the initial reconstruction image block of the cooperative reconstruction group $Z_i$, as shown in Eq. (3).

$$Z_i = F_1(W_1, Y_i), i \in [1, N] \tag{3}$$

where $F_1$ represents the fully connected layer. $W_1$ is the weight of the fully connected layer, which is obtained from the network training. F uses the fully connected network layer $F_1$ to upgrade the dimension and convert the compressed observed value in the collaborative reconstruction group $Y_i$, and obtains the initial reconstructed image block collaborative reconstruction group with the size of $B \times B$.

The channel stitching technique is used to fuse the non-local similar features of the image. The initial reconstruction block co-reconstruction group $Z_i$ contained in the

initial reconstruction estimates of the image blocks $z_i$ and its initial estimate of non-local similar features $z_{i,1}, z_{i,2}, \ldots, z_{i,5}$. The joint channel stitching is performed to obtain the co-reconstructed features with fused non-local similar features $Z_c$, whose equation is shown in Eq. (4).

$$Z_c = \mathrm{concat}\left(z_i, z_{i,1}, \ldots, z_{i,5}\right) \tag{4}$$

where concat means channel splicing.

Unlike traditional convolutional operations focusing on local neighborhoods, this module aggregates information from all positions, enriching the feature representations with a global context. It operates through non-local operations that compute responses at each position as a weighted sum of features from all other positions, with weights determined adaptively based on feature similarity. This process involves generating an attention map highlighting the significance of different feature positions, guiding the aggregation of features to focus on the most relevant parts of the input.

First, the reconstruction network uses a fully connected layer to increase the dimension of the input measurements and reconstruct them into smaller feature images. Then, a sub-pixel convolution module composed of a sub-pixel convolution layer, a 3×3 convolution layer, and a batch normalization layer is used to improve the image discrimination rate gradually, and the calculation process is shown in Eq. (5).

$$I^{HR} = f^L\left(I^{LR}\right) = \mathrm{PS}\left(W_L * f^{L-1}\left(I^{LR}\right) + b_L\right) \tag{5}$$

where $I^{LR}$ is the low-resolution image, that is, the smaller feature map. $I^{HR}$ is the reconstructed high-resolution image, which is the final reconstructed image. $f^L$ is the output image of layer $L$, and the calculation process of PS function is shown in Eq. (6).

$$\mathrm{PS}(T)_{x,y,c} = T_{\lfloor x/y \rfloor \lfloor y/r \rfloor, c \cdot r \cdot \mathrm{mod}(y,r) + c \cdot \mathrm{mod}(x,y)} \tag{6}$$

where PS operation can transform the low-resolution image with size $H \times W \times C \cdot r^2$ into the high-resolution image with size $rH \times rW \times C$ by periodic transformation.

The reconstruction process of the reconstructed image $\widetilde{x}$ (Eq. 7) shows where $R$ is the depth reconstruction network.

$$\widetilde{x} = R\left(f^L(y)\right) = \mathfrak{PS}\left(f^{L-1}(y)\right). \tag{7}$$

## Enhanced reconstruction based on global residual module

The concept of residual networks proves to be highly applicable in the image reconstruction phase, and this study introduces a novel approach by proposing a fully convolutional residual block. During the network training process, a crucial mechanism is employed wherein input image features are directly connected across layers to the output. This connectivity preserves the convolution-extracted image features and prevents their loss in the forward channel as the network architecture deepens. This approach accelerates the convergence of the network by maintaining the integrity of essential image features throughout the reconstruction process.

Simultaneously, the loss function is crucial in optimizing the image reconstruction effect. By comparing the residual information of the input image with each generated image during training, the loss function progressively minimizes the disparity between them. This iterative refinement process allows the network to continuously improve its ability to reconstruct images accurately, ultimately enhancing the overall quality of the reconstructed images.

By leveraging the power of residual networks, incorporating the novel full convolutional residual block, and effectively utilizing the loss function, this proposed approach effectively addresses the challenges associated with image reconstruction. Preserving image features, accelerated network convergence, and iterative refinement all contribute to achieving superior image reconstruction results. Since the image obtained in the initial reconstruction process is degraded compared to the original image, the image blocks are refined and reconstructed. The obtained cooperative joint reconstruction features are $Z_c$ into a global residual reconstruction network consisting of non-local feature adaptive interaction modules $F_{lg}$ global residual reconstruction network composed of a stack of non-local feature adaptive interaction modules $F_r$. The final output image is obtained by fusing the features $z_i'$ as shown in Eq. (8):

$$z_i' = z_i + F_r(W_2, Z_c) \qquad (8)$$

where $z_i$ is the initial estimate of the image block, $W_2$ is the residual network parameter.

The concept of residual networks is leveraged in the image reconstruction stage, where a state-of-the-art full convolutional residual block is proposed. During network training, input image features are directly connected across layers to the output, thereby ensuring the preservation of features extracted through convolution. This approach efficiently circumvents the deepening of network structure that can result in the loss of image features in the forward channel, thus accelerating network convergence. Additionally, the loss function is utilized to compare the residual information of the input image with each training-generated image, continually decreasing the differences between them and ultimately enhancing the image reconstruction effect.

Figure 3 illustrates that the residual block comprises a feedforward network featuring three convolutional layers, accompanied by a constant jump link. where x denotes the input feature map, the residual function is denoted by F(x), and the output of the residual block is denoted by H(x). Therefore, the expression of the residual block is shown in Eq. (9).

$$H(x) = F(x) + x. \qquad (9)$$

When F(x) tends to zero, which is equivalent to the output of the residual block H(x) and the input x is extremely close to the input, and the difference is small, *i.e.,* H(x) = x . This is called the constant mapping of the residual function; after training the network with many data sets, the optimal value of F(x). The optimal value of the residual block is obtained as the output of the residual block H(x). The residual block's output is acquired after training the network on many data sets. During the backpropagation stage, the gradients utilized for training the three convolutional layers within the residual block are saved. Additionally,

**Peer**J Computer Science

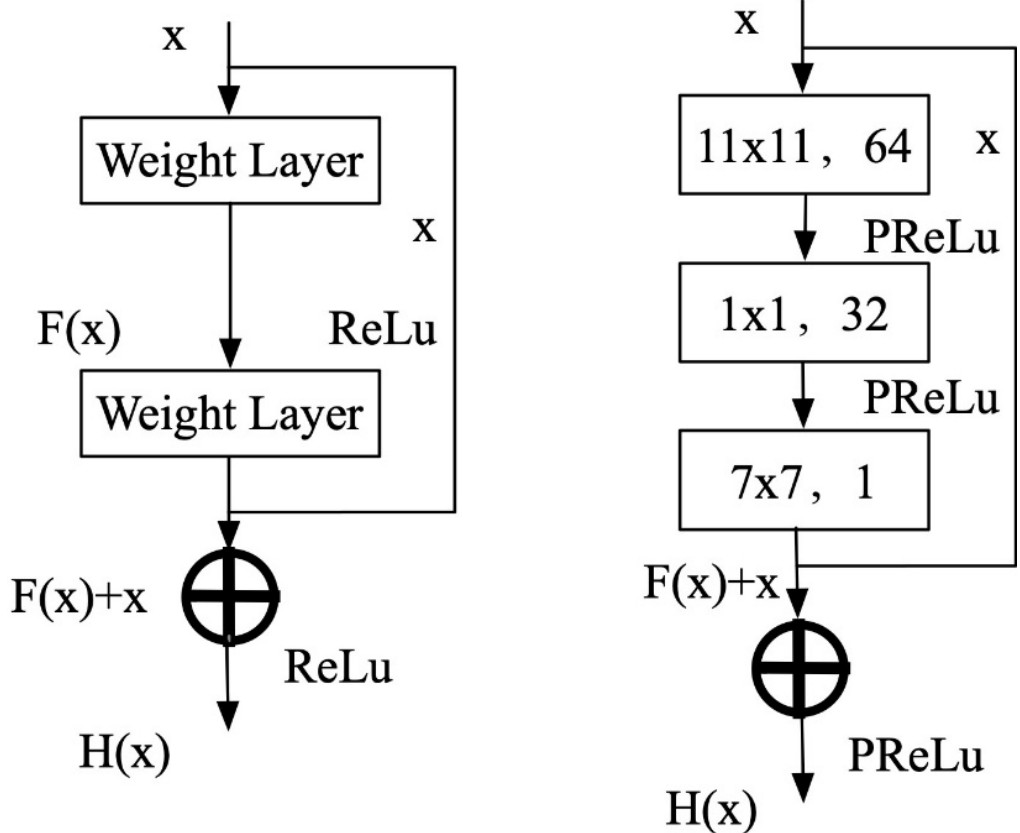

**Figure 3** Residual block structure diagram.

each convolutional layer is succeeded by a PReLU activation function, thereby augmenting the network model's non-linear expression.

## Loss function

To begin with, these images must be initialized by converting them from RGB space to YCrCb. Subsequently, the image space is transformed from RGB to image space, with the luminance channel being selectively chosen. The network is then trained with four distinct sampling rates: 1%, 4%, 10%, and 25%, correspondingly. Diverging from conventional approaches in the domain of compression-aware image reconstruction, the proposed network is jointly trained alongside a complete convolutional residual reconstruction sub-network, with the loss function defined by Eq. (10):

$$L(W) = \frac{1}{T} \sum_{i}^{T} \left\| f(x_i, W, K) - x_i \right\|^2 \tag{10}$$

where T denotes the number of channels of the feature map during network training, W denotes the material weight parameter in the pick-and-place network, K denotes the weight parameters in the full convolutional residual reconstruction sub-network, and the two parts

are jointly trained in a unified way to train the whole network by back-propagating the Euclidean distance between the original image labels and the reconstructed images.

# EXPERIMENT AND ANALYSIS

## Experimental setup

The experiment employs the 91-images dataset (https://zenodo.org/records/11163729, 10.5281/zenodo.11163729) and the BSD300-train dataset (https://zenodo.org/records/10667264, 10.5281/zenodo.10667264) as the training dataset. The 1-dimensional column vector is normalized in the range (0,1) for each dimension to expedite network convergence. The input to the network is obtained by sampling image blocks cropped from each image using a Gaussian random matrix, with the luminance components of the image blocks being extracted as supervised labels during training. The observation matrix employed in this experiment comprises a Gaussian random matrix that satisfies finite isometric constraints, with the sampling rates set to {0.01, 0.04, 0.10, 0.25}. The model hyperparameters are presented in Table 1.

To assess the efficacy of diverse algorithms, grayscale images extracted from the BSD200-train dataset were utilized for testing, a Jupyter notebook with eight different solutions for common problems of digital image processing, including object recognition and binarization using an adaptive threshold. The evaluation metrics were peak signal-to-noise ratio (PSNR) and structural similarity index (SSIM).

The experimental environment utilized in this study is detailed in Table 2.

## Analysis of results

Figure 4 visually depicts the mean PSNR values obtained from the test images. Initially, at a learning rate of 0.001, the PSNR curve shows fluctuations as the epochs progress. This fluctuation can be attributed to the learning process as the model gradually adapts to the data. However, the curve exhibits a more consistent pattern as the learning rate decays. This signifies that the model is converging and reaching a stable state, where further training iterations have minimal impact on improving the PSNR values. The plateauing of the curve indicates that the trained model has achieved convergence across the four distinct sampling rates, demonstrating the effectiveness of the learning rate decay strategy in facilitating model stability and optimizing reconstruction quality.

To assess the effectiveness of the proposed network, a comprehensive comparison was made with five representative deep learning-based CS techniques, namely ReconNet, NL-MRN, ISTA-Net, SCSNet, and GMNet. The evaluation of these algorithms involved utilizing nine standard grayscale images, which served as the test bed for the analysis. The resulting performance metrics and findings are graphically represented in Figs. 5 and 6, providing a clear visual representation of the comparative outcomes achieved by each technique.

The proposed model demonstrates remarkable performance, surpassing other deep learning-based techniques, as substantiated by its consistently highest average PSNR and SSIM values across the four sampling rates. SSIM, being a more sophisticated image similarity evaluation index, provides deeper insights into the quality of the reconstructed

**Table 1  Hyperparameter settings.**

| Name | Hyperparameter value |
| --- | --- |
| Optimizer | Adam |
| Learning rate | 0.01 |
| lr_policy | Multistep |
| Batch size | 10 |
| Epoch | 50 |
| Momentum | 0.5 |
| Learning rate change ratio | 0.9 |

**Table 2  The experiment environment.**

| Environment | Information |
| --- | --- |
| CPU | I7-8750HQ |
| GPUs | GTX 1080 |
| Language | Python 3.5 |
| Framework | Tensor flow and Scikit-learn |

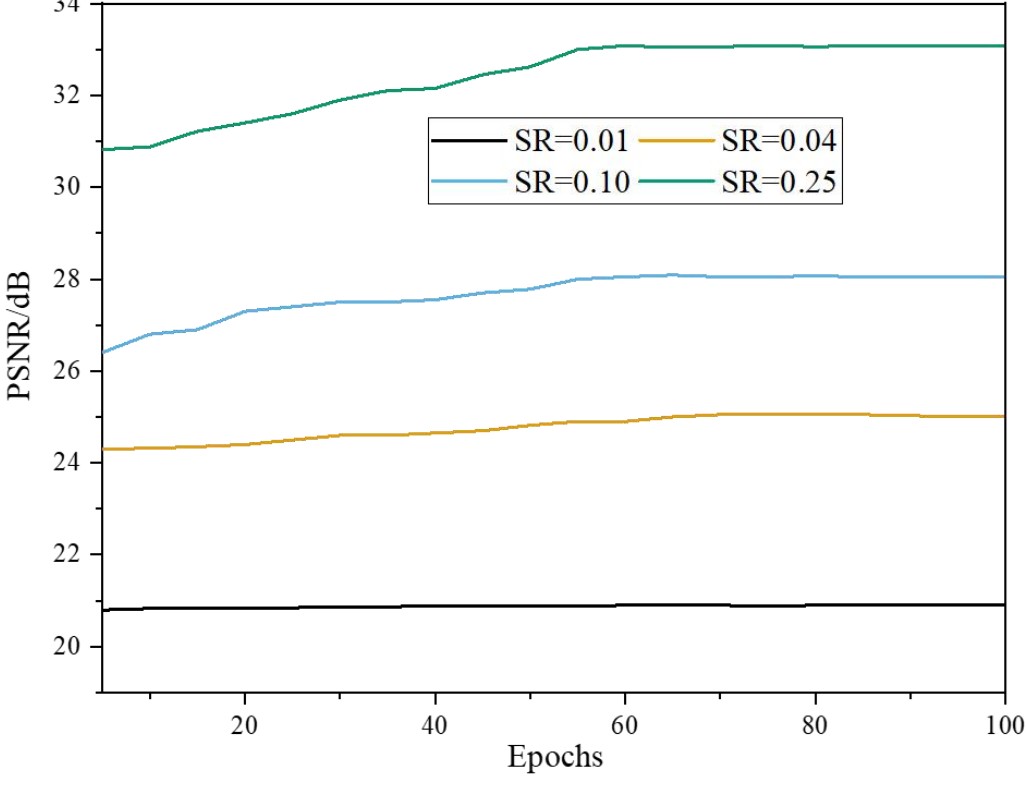

**Figure 4  Results of PSNR of the model with Epochs under the BSD200 dataset.**

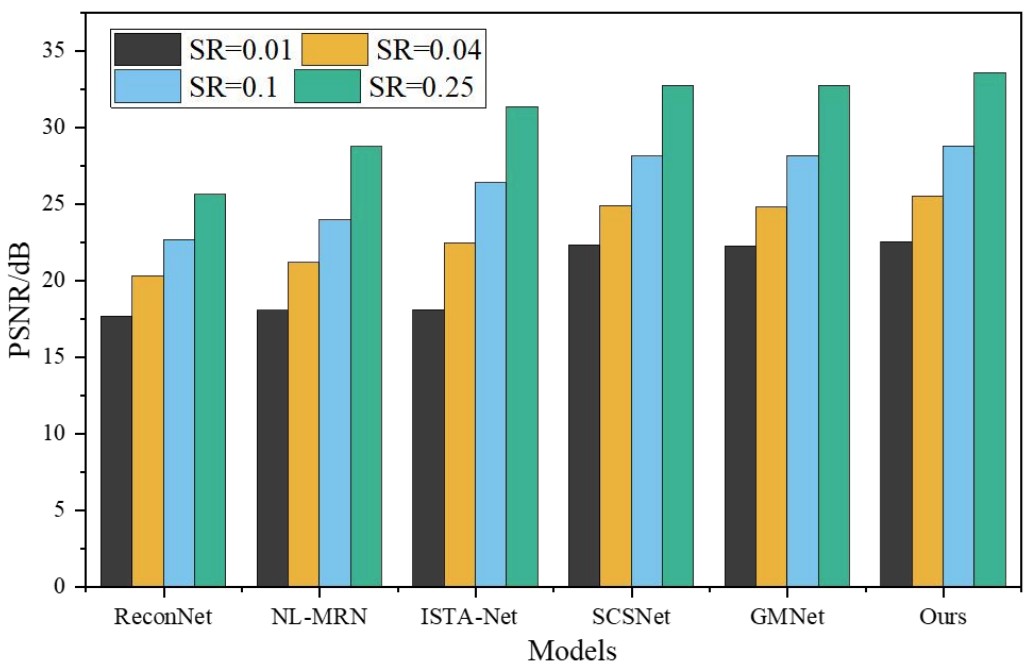

**Figure 5** PSNR under different models and SR.

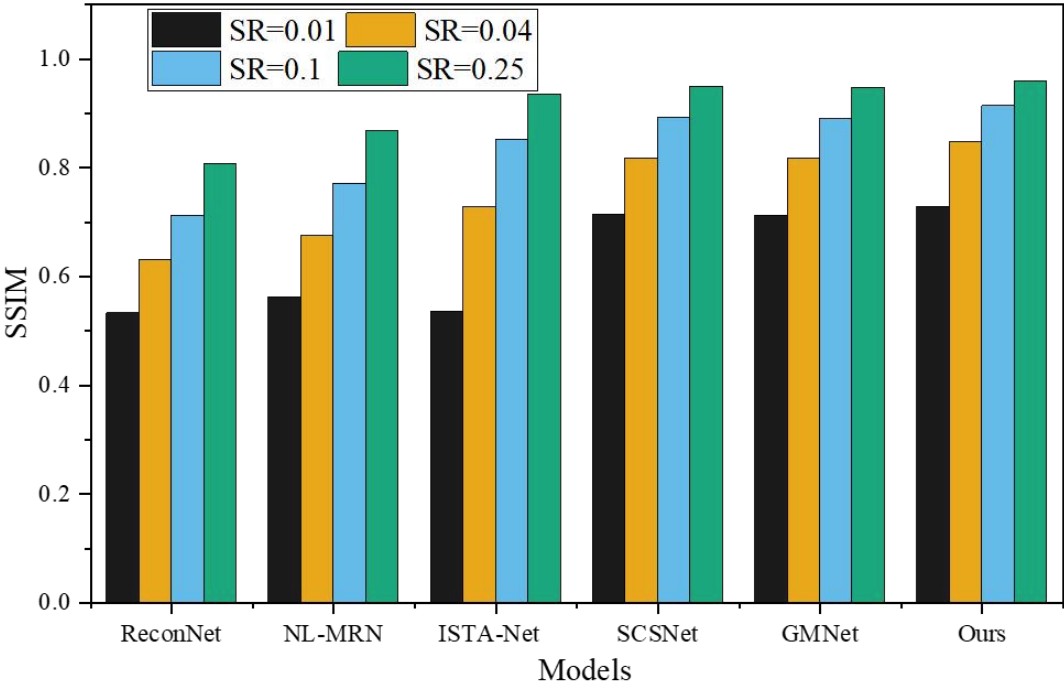

**Figure 6** SSIM under different models and SR.

images. Notably, the network architecture introduced in this study showcases superior SSIM results compared to the reference algorithms. When SR is 0.01, 0.04, 0.10, 0.25, the average PSNR is 4.87dB, 5.23dB, 6.12dB, 7.95Db higher than ReconNet. The average improvement over GMNet is 0.27dB, 0.74dB, 0.64dB and 0.76dB, showing superior reconstruction performance.

This achievement can be attributed to the novel non-local feature adaptive interaction module incorporated within the proposed model. By integrating non-local feature fusion convolution, this module effectively combines non-local information, allowing for comprehensive and accurate fusion of features. Moreover, it introduces weight adjustments to highlight critical high-frequency features while diminishing the influence of redundant low-frequency data. This is accomplished by utilizing the channel correlation discriminant and spatial correlation discriminant modules, enabling a more refined and discriminating feature representation.

Furthermore, the proposed model offers a unique advantage by doubly suppressing features that exhibit in-group matching errors. By identifying and suppressing these erroneous features, the model significantly enhances image reconstruction accuracy. This suppression mechanism adds layer of refinement to the reconstruction process, resulting in improved image quality and fidelity.

The proposed model outperforms existing deep learning-based techniques, as evidenced by its superior average PSNR and SSIM values across various sampling rates. The introduction of the non-local feature adaptive interaction module, coupled with the suppression of in-group matching errors, enables the model to achieve exceptional reconstruction accuracy. These advancements generate high-quality reconstructed images, solidifying the efficacy of the proposed network architecture for image reconstruction tasks.

In addition to evaluating the image reconstruction quality of different techniques, it is equally important to consider comparing their respective reconstruction times. Figure 7 provides insights into the average GPU runtime necessary for reconstructing nine images across various sampling rates, each with a size of 256 × 256.

It is worth noting that Fig. 7 exclusively focuses on the time consumed by the network during the image reconstruction process to ensure a fair and accurate comparison. By analyzing the GPU runtime, the author comprehensively understands the computational efficiency and speed of the different techniques, allowing the author to assess their suitability for real-time applications or scenarios with time constraints.

Analyzing Fig. 7, it becomes discernible that ReconNet exhibits a lower computational complexity than the other techniques. However, its reconstruction performance falls short regarding both PSNR and visual effects. On the other hand, NL-MRN showcases slightly improved reconstruction performance, albeit with a marginally longer reconstruction time compared to ReconNet.

Another noteworthy technique is GMNet, which achieves superior reconstruction quality while retaining the fastest reconstruction speed among the evaluated methods. Combining high-quality results and efficient processing makes GMNet a compelling option for image reconstruction tasks.
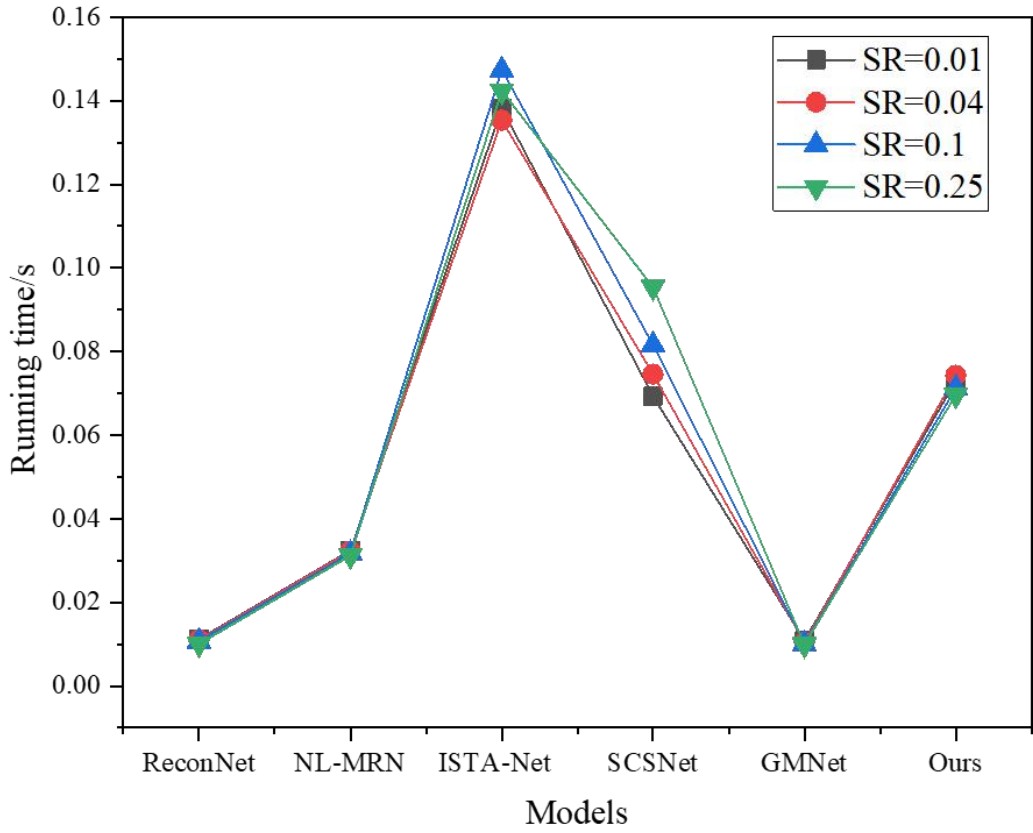

**Figure 7** Running time under different models and SR.

Furthermore, the proposed model is evaluated against SCSNet. Notably, at low sampling rates, the reconstruction time of SCSNet is nearly on par with that of the proposed model. However, as the sampling rate increases, the depth of the SCSNet network grows, resulting in a corresponding increase in reconstruction time.

Comparatively, ISTA-Net exhibits the longest image reconstruction time while delivering average reconstruction quality. In summary, compared to these deep learning-based image reconstruction methods, the proposed model outperforms the others in terms of image reconstruction time and overall reconstruction performance. Its superior balance of efficient processing and high-quality results make it a promising choice for various image reconstruction applications.

## CONCLUSION

Numerous types of graphic design software are available, with the most common applications being dot matrix images and vector graphics, sharing similarities and differences. Dot matrix images have a fixed color resolution and may be inadequately reconstructed due to insufficient feature extraction, resulting in poorly reconstructed edge contours and detailed textures. This article proposes a non-local feature fusion network-based image reconstruction method. Experimental results demonstrate that the

proposed method accurately estimates the original image, with clearly visible texture details compared to previous image CS reconstruction methods. The potential of our proposed model lies in its ability to effectively reconstruct high-frequency information, which is critical for maintaining the fidelity of textures and edges in graphic design applications. High-frequency information is essential for precisely rendering fine details, sharp edges, and intricate textures, crucial for creating visually appealing and professional-quality designs. By accurately capturing and reconstructing these high-frequency components, our method ensures that the final images retain their original clarity and detail, making them highly valuable for graphic design tasks that demand high precision and quality. Furthermore, the proposed method's ability to preserve detailed textures and edge contours significantly enhances the functionality of widely used graphic design software, such as Photoshop, for tasks involving image refinement and compositing. Ensuring reasonable shadows, saturation, and other detailed aspects of an image is critical for creating realistic and compelling visual content.

Notably, a set of generators and discriminators may produce texture-rich images from random noise. Moreover, compressed observations containing most of the original image's information may be reconstructed using generative adversarial networks. Typically, detected edges are coarse, leading to insufficient reconstruction of the reconstructed image's edge information. Hence, improving edge accuracy is essential. Our method addresses this issue by incorporating advanced feature fusion techniques, resulting in superior edge reconstruction and overall image quality, making it a robust solution for graphic design applications.

### Funding
The authors received no funding for this work.

### Competing Interests
The authors declare there are no competing interests.

### Author Contributions
- Xinxin Fu conceived and designed the experiments, performed the experiments, performed the computation work, prepared figures and/or tables, authored or reviewed drafts of the article, and approved the final draft.
- Lujing Tang conceived and designed the experiments, analyzed the data, performed the computation work, prepared figures and/or tables, and approved the final draft.
- Yingjie Bai conceived and designed the experiments, performed the experiments, analyzed the data, performed the computation work, authored or reviewed drafts of the article, and approved the final draft.

### Data Availability
The BraggSpotFinder dataset is available at Zenodo: Jakoncic, J. (2024). BraggSpotFinder Data Set (BSD) [Data set]. Zenodo. https://doi.org/10.5281/zenodo.10667264.

The 91-images dataset is available at Zenodo: None. (2024). 91 images [Data set]. Zenodo. https://doi.org/10.5281/zenodo.11163729.

## Supplemental Information

Supplemental information for this article can be found online at http://dx.doi.org/10.7717/peerj-cs.2227#supplemental-information.

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
