# Peer review of "Image reconstruction in graphic design based on Global residual Network optimized compressed sensing model"

_PeerJ Computer Science, doi:10.7717/peerj-cs.2227_

## Round 0.1 · original submission · Major Revisions

Dear author/s
Thank you for submitting your work in our esteemed journal. Your paper has been reviewed with interest by the experts in the field. You will see that their comments are necessary to enhance the quality of your manuscript. Therefore, We must send you a request to revise your article in light of these comments, alongside, please also respond/incorporate the following comments in your revised paper.

The abstract is detailed but could benefit from a more concise summary of the methodology and key findings. Highlight the novel aspects and significance of the work in fewer sentences.

Provide a more detailed explanation of the channel stitching process. Visual aids or diagrams could help clarify this step.

Clearly define the metrics used to evaluate image reconstruction accuracy. Explain why these metrics are appropriate for assessing the performance in graphic design contexts.

Suggest directions for future research. Consider how the methodology could be expanded or applied to other areas of image processing

Reviewer 1 ·

Basic reporting

The authors in this article study the optimization of the traditional Compressed Sensing model to address information degradation and distortion in graphic design. This model creates a co-reconstruction group from compressed observations, reconstructs the images, and inserts it into a global residual network. The results show low sampling rates and effective reconstruction of high-frequency information, meeting the specific requirements of image processing in graphic design.
The study is well structured and the study of literature is appropriate to the topic covered. The model is robust and well defined.
There are some aspects that need to be worked on to improve the papar.
1. The authors write that the graphic design work is carried out with the help of software, but the type of software used is not specified. Therefore, it is recommended to put it in a note.
2. Figure 2 is not readable. It is recommended to improve its resolution.
3. Careful revision of the English is recommended for the entire document.

Experimental design

4. In section 4.1, the use of the set6 dataset for testing is introduced. It is recommended to provide more information on the creation of the set6 dataset.
5. Additionally, it is recommended that the preprocessing techniques used for the dataset cleaning phase are listed before it is used for the training phases.
6. Section 4.1 indicates the performance of the computer on which the experiments were conducted but the software used to carry out the experimental analysis is not clearly shown

Validity of the findings

The results are well defined and clear strengthened by comparison with other methods.
7. It is recommended to better explain the potential of the model used, in particular, how the proposed method effectively reconstructs high-frequency information and why this is important for graphic design applications.

Reviewer 2 ·

Basic reporting

The manuscript addresses a significant problem in the field of graphic design and proposes an innovative method for image reconstruction. However, the explanation of the methodology needs to be more detailed, and the experimental results should be presented with comprehensive quantitative analysis. Addressing these issues will enhance the clarity and impact of the research.

Experimental design

The introduction should clearly articulate the problem of information degradation and distortion in graphic design, providing more context about why this problem is significant and how it impacts the field.

More specifics are needed on the initial reconstruction process within similar groups. Elaborate on the techniques used for this initial step and how it contributes to the overall reconstruction quality.

The process of channel stitching should be explained in greater detail. Describe how this step integrates with the global residual network and its role in enhancing local feature reconstruction.

The structure and functioning of the global residual network need to be described more comprehensively. Include details about the non-local feature adaptive interaction module and how it aids in feature fusion.

Provide a rationale for choosing the 91-images dataset and BSD200-train dataset for training and the set6 dataset for testing. Explain why these datasets are appropriate for the task and any preprocessing steps taken.

Validity of the findings

Include detailed information on the experimental setup, including the parameters used for training the model. This should cover learning rates, batch sizes, and any other relevant hyperparameters.

The results should be presented with more quantitative data, including metrics such as PSNR (Peak Signal-to-Noise Ratio) and SSIM (Structural Similarity Index) to support claims of improved image reconstruction accuracy.

Additional comments

Discuss in more depth the techniques used to improve edge accuracy and how they specifically address the coarse detected edges seen in previous methods.

---

## Round 0.2 · accepted · Accept

Dear Authors,

I am pleased to inform you that the reviewers are now satisfied with the revised version of your manuscript and they are recommending it for publication. Therefore, we are pleased to let you know about the acceptance of your article.

thank you

Reviewer 1 ·

Basic reporting

no comment

Experimental design

no comment

Validity of the findings

no comment

Reviewer 2 ·

Basic reporting

All the suggestions have been incorporated.

Experimental design

Ok

Validity of the findings

Ok

Additional comments

NA